# Improved Mechanical, Anti-UV Irradiation, and Imparted Luminescence Properties of Cyanate Ester Resin/Unzipped Multiwalled Carbon Nanotubes/Europium Nanocomposites

**DOI:** 10.3390/ma14154244

**Published:** 2021-07-29

**Authors:** Na Yang, Xiaohua Qi, Di Yang, Mengyao Chen, Yao Wang, Linjun Huang, Olga Grygoryeva, Peter Strizhak, Alexander Fainleib, Jianguo Tang

**Affiliations:** 1Institute of Hybrid Materials, National Centre of International Joint Research for Hybrid Materials Technology, National Base of International Sci. & Tech. Cooperation on Hybrid Materials, Qingdao University, 308 Ningxia Road, Qingdao 266071, China; 17669490805@163.com (N.Y.); qixiaohua0930@163.com (X.Q.); yd130102@163.com (D.Y.); chenmengyao210@163.com (M.C.); wangyaoqdu@126.com (Y.W.); huanglinjun@qdu.edu.cn (L.H.); 2Institute of Macromolecular Chemistry of the National Academy of Sciences of Ukraine, 02068 Kyiv, Ukraine; grigoryevaolga@i.ua (O.G.); fainleib@i.ua (A.F.); 3B.L.V. Pysarzhevskii Institute of Physical Chemistry, National Academy of Sciences of Ukraine, 31 Prosp. Nauky, 03028 Kyiv, Ukraine; pstrizhak@hotmail.com

**Keywords:** cyanate ester resin, unzipped multiwalled carbon nanotubes, europium, nanocomposites, structure–property relationships, ultraviolet aging, luminescence

## Abstract

Cyanate ester resin (CER) is an excellent thermal stable polymer. However, its mechanical properties are not appropriate for its application, with brittle weakness, and it has poor functional properties, such as luminescence. This work innovatively combines the luminescence property and the improved mechanical properties with the inherent thermal property of cyanate ester. A novel nanocomposite, CER/uMWCNTs/Eu, with multi-functional properties, has been prepared. The results show that with the addition of 0.1 wt.% of uMWCNTs to the resin, the flexural strength and tensile strength increased 59.3% and 49.3%, respectively. As the curing process of the CER progresses, the injected luminescence signal becomes luminescence behind the visible (FBV). The luminescence intensity of CER/uMWCNTs/Eu was much stronger than that of CER/MWCNTs/Eu, and the luminescence lifetime of CER/MWCNTs/Eu and CER/uMWCNTs/Eu was 8.61 μs and 186.39 μs, respectively. FBV exhibited great potential in the embedment of photon quantum information. Therefore, it can be predicted that CER/uMWCNTs/Eu composites will not only have a wide range of applications in sensing, detection, and other aspects, but will also exhibit great potential in the embedding of photon quantum information.

## 1. Introduction

Cyanate ester resin (CER) is an excellent thermal stable polymer, with a unique triazine ring network structure, which can be used at high temperatures for a long time. Therefore, it is widely applied in various fields, such as aerospace, microelectronics packaging field, and so on [1,2,3,4]. However, its mechanical properties are not appropriate for its application, with brittle weakness, and it has poor functional properties, such as luminescence [5,6,7]. Therefore, it is important to find an additive that can not only improve its mechanical properties such as toughness, but also increase its functional properties.

Multiwalled carbon nanotubes (MWCNTs) have excellent structural characteristics, are easy to functionalize, and can be used as reinforcement materials to improve the performance of polymer materials [8,9,10,11]. CER/MWCNTs nanocomposites have been recently developed and studied by several teams [12,13,14,15,16,17,18,19,20]. The catalytic effect of MWCNTs on CER polymerization was fixed, and CER/MWCNTs nanocomposites with improved mechanical and thermal properties and electrical and thermal conductivity have been elaborated. However, the matrix reinforcement of MWCNTs is not significant because the contact area between the carbon nanotubes and the matrix is limited, and the outermost nanotubes prevent the inner tubes from contacting the polymer matrix. [21]. Kosynkin et al. reported a representative way to form unzipped multiwalled carbon nanotubes (uMWCNTs) [22]. Since then, many reports have begun to use uMWCNTs as fillers to strengthen polymer composites [23,24]. Those demonstrated that uMWCNTs have outstanding physical enhancement properties and are superior to those of MWCNTs [25,26]. As a filler, carbon nanotubes can not only improve the brittleness of polymer, but also functionalize polymer materials.

Cyanate resins are commonly used in aerospace applications, hence UV aging resistance is also an important indicator to measure their performance [27]. In addition, the material can also be damaged by ultraviolet radiation and other environmental factors in their actual use, which will cause aging of the material. The aging of polymer material is an important factor affecting the safety of a sealed cabin, which requires it to have the ability to resist solar radiation, the thermal cycle, electronic radiation, and other factors in space [28]. Cyanate ester resin itself has excellent UV aging resistance. The effect of uMWCNTs on the UV aging performance of cyanate resin was also studied in this paper.

Furthermore, we also injected luminescence properties into the cyanate ester resin matrix. Just a few publications have reported the combination of the luminescent properties of lanthanide complexes with heat-resistant polymers [29,30,31]. We have innovatively combined lanthanide complexes with CER to obtain light-emitting and heat-resistant polymer composites. After curing, the luminescence emission spectrum of resin usually has a peak at 500 nm, which is a characteristic peak of polymer materials containing aromatic rings. However, the new luminescence signal of the lanthanide complex we injected is different from the original peak of the resin. With the formation of a cyanate ester network during curing, the injected luminescence signal changed to luminescence behind the visible (FBV). FBV has great application potential in the embedment of photon quantum information.

As a kind of special luminescence materials, lanthanide complexes are often combined with various organic or inorganic materials to prepare hybrid materials for practical use. In recent years, polymers, as a kind of material with excellent mechanical properties, are also combined with lanthanide complexes. However, pure lanthanide complexes and polymer matrix have unstable luminescence properties. Therefore, we first attached the lanthanide compound on the side wall of carbon nanotubes to prepare luminescence carbon nanotubes [32,33]. For example, Wu et al. coated the multiwalled carbon nanotubes by lanthanide (Eu^3+^, Tb^3+^) complexes and obtained luminescence materials with excellent luminescence stability through a simple in situ synthesis method [34]. In addition, functional groups on a surface of carbon nanotubes can also enhance the interface interaction with polymer [35]. Thus, excellent mechanical and processing properties of the matrix and luminescence of lanthanide complex are combined to obtain a new type of luminescent materials with excellent mechanical and luminescence properties.

Herein, a novel kind of cyanate ester resin/unzipped multiwalled carbon nanotubes (uMWCNTs) luminescent nanocomposite was developed, which combined the excellent mechanical properties of heat resistance, UV aging resistance of the polymer nanocomposite and the luminescence properties of the lanthanide complex. First, the enhancement effect of uMWCNTs on the toughness and thermal properties of CER was determined. On this basis, we added luminescence signal to the composite material, resulting in the composite with unique combination of toughness, thermal, and luminescence properties. It could be predicted that thermally resistant polymer hybrid luminescent materials would be widely used in sensing, detection, and hazard prediction.

## 2. Materials and Methods

### 2.1. Materials

The cyanate ester resin monomer (Bisphenol A dicyanate ester, DCBA) was supplied from Kaixin New Material Technology Co., Ltd. (Liyang, China). Multiwalled carbon nanotubes (MWCNTs, length: 0.5–2 μm, diameter: 50 nm) were bought from Xianfeng Nano Material Technology Co., Ltd. (Nanjing, China). EuCl_3_ (99.99%), TTA (99%), and phen (99.99%) were procured from Maclean Biochemical Technology Co., Ltd. (Shanghai, China). Acetone, H_2_SO_4_ (98%), H_2_O_2_ (30%), HCl (36%) and KMnO_4_ were also received from Maclean Biochemical Technology Co., Ltd (Shanghai, China).

### 2.2. Methods

#### 2.2.1. Preparation of Unzipped Multiwalled Carbon Nanotubes (uMWCNTs) and MWCNTs/Eu and uMWCNTs/Eu Composites

First, 0.2 g of MWCNTs were dispersed in 35 mL of H_2_SO_4_ and sonicated for at least 10 h. Then, 1 g KMnO_4_ was added in several times under magnetic stirring, and then raised the temperature to 50 °C, and held for 1 h. After that, quickly poured the mixture into a beaker filled with H_2_O_2_ ice. If bubbles were generated, added 0.2 mL of H_2_O_2_ solution and repeated the step of adding H_2_O_2_ solution multiple times until no bubbles were generated. Then, the mixed solution was centrifuged and washed once with dilute HCl (10%) and deionized water several times until the pH of the mixed solution was neutral, and then dried in oven at 60 °C. The resulting black powder was unzipped multiwalled carbon nanotubes, coded as uMWCNTs.

Next, the MWCNTs/Eu and uMWCNTs/Eu composites were prepared. Appropriate amounts of MWCNTs or uMWCNTs were uniformly dispersed in ethanol at 60 °C. Then, the phen, TTA and EuCl_3_ were successively added and stirred for 2 h [36]. The pH value was adjusted to 11, and the luminescence intensity was the highest when the pH value was 11 [37]. The mixture was washed to remove excess Eu(TTA)_3_phen complexes and dried. The uMWCNTs/Eu composites, which had a europium complex on the surface of uMWCNTs, have been obtained.

#### 2.2.2. Preparation of CER/uMWCNTs and CER/uMWCNTs/Eu Nanocomposites

uMWCNTs were dispersed (1 mg/mL) by ultrasonication in acetone at room temperature for 30 min. Then CE monomer added to the above acetone solution and stirred vigorously for 2 h at 100 °C [38]. The mixed solution was dried in an oven for 24 h at 80 °C to remove acetone and cast into a preheated mold coated for curing. The CER/ uMWCNTs nanocomposite was synthesized using step by step curing schedule consisted of the following stages: 1 h at 120 °C, 1 h at 140 °C, 2 h at 160 °C, 2 h at 180 °C, 2 h at 200 °C, 4 h at 230 °C and post curing via 1 h at 250 °C [39]. The concentration of uMWCNTs was 0.1 wt.%. The preparation method used for CER/uMWCNTs nanocomposite synthesis is schematically shown in Figure 1.

The CER/uMWCNTs/Eu nanocomposite containing 0.1 wt.% of uMWCNTs/Eu was prepared similarly.

#### 2.2.3. UV Irradiation Exposure

In the process of actual use, the material will be affected by ultraviolet radiation and other environmental factors, which will cause the aging of the material. Samples of the nanocomposites were exposed to UV radiation at 50 °C in an accelerated climate chamber which was self-made in the laboratory, equipped by four rows of UV lamps with a UV radiation wavelength of 340 nm each and the intensity was 0.71 W/m^2^ [40]. The total dose of UV irradiation applied was 9.6 × 107 J/m^2^ (40 days of exposure). The sample was 5 cm away from the UV lamp and placed in parallel.

### 2.3. Characterization

Fourier transformed infrared spectroscopy (FTIR, MAGNA-IR 5700) (Thermo Nicolet Corporation, Waltham, MA, USA), X-ray diffraction (XRD, D8 Advance) (Bruker, Karlsruhe, Germany), Raman (Almega Thermo Nicolet) (Thermo Nicolet Corporation, Waltham, MA, USA) and X-ray photoelectron spectroscopy (XPS, ESCALAB-210) (Thermo Fisher Scientific, Waltham, MA, USA) were applied to characterize the structure of carbon nanotubes.

The cross section of the nanocomposites was characterized by scanning electron microscope (SEM, JEOL 6460) (JEOJ, Kyoto, Japan). The MWCNTs and unzipped MWCNTs were characterized using Transmission Electron Microscopy (TEM, JEM-F2100) (JEOJ, Kyoto, Japan). The outer diameter of MWCNTs and uMWCNTs were measured using Nano Measurer software (Nano Measurer 1.2, Shanghai, China). The number of nanotubes of each type was equal, and at least 50 nanotubes were measured. The flexural and tensile properties of the composites were measured with a universal testing machine (CMT4304GD) (MTS, Jinan, China); Sample size was 75 mm × 5 mm × 4 mm, and five samples were measured for each nanocomposite. The thermal behavior was characterized using Differential Scanning Calorimeter (DSC, DSC214 Polyma) (NETZSCH, Cologne, Germany) and Thermogravimetric analysis (TGA, SDT Q600) (TA Instruments, New Castle, PA, USA). The initial degradation temperature (T_di_) was determined as the 5% weight loss temperature. Luminescence intensity and lifetime were recorded on the were recorded on Edinburgh luminescence spectrofluorometer (FLS980-STM) (Edinburgh Instruments, Edinburgh, United Kingdom).

## 3. Results and Discussion

### 3.1. Structure of uMWCNTs, MWCNTs/Eu and uMWCNTs/Eu

Unzipped multiwalled carbon nanotubes (uMWCNTs) is a portrayal description in references/publications [22], which indicates the geometric change of CNTs. In this unzipping process, the oxidation of pristine CNTs is the chemical cause to initiate this morphological structural change of MWCNTs. It should be noted that in this process, the one-dimensional MWCNTs are destroyed and changed into open morphological structures with different geometric shapes, as shown in Figure 2. The TEM images of pristine MWCNTs and unzipped MWCNTs show that the morphology of the carbon nanotubes changed significantly after their unzipping. As shown in Figure 2, the pristine MWCNTs can be seen with smooth outer contours and the tubular structure is obvious. Compared with pristine MWCNTs, the length of uMWCNTs became shorter after strong oxidation method processing, and the outer walls were partially loosened or unwrapped, or even fell off from the surface of the nanotubes to form graphene nanosheets, thus increasing the surface area of MWCNTs. Instead of forming flat graphene sheets, however, they show a tubelike structure. The size distribution of the outer width for the MWCNTs and uMWCNTs in the diagrams inside the corresponding TEM images also illustrated this phenomenon. Figure 2 shows that the pristine MWCNTs have inherent hollow tubular structure with the average outer width of D = 42.68 ± 5.06 nm, while the average outer width of uMWCNTs is D = 83.0 ± 1.53 nm. One can see that the width of uMWCNTs is almost twice compared to that of MWCNTs, indicating effective unzipping [22]. The increase of the width of carbon nanostructures not only provided a larger surface area for the binding of polymer resin to carbon nanotubes, but also increased the number of groups available on the edge and surface of MWCNTs for further improvement of effective matrix/filler interaction and the dispersion stability.

Figure 3a illustrates the surface structure of MWCNTs characterized by FTIR. The FTIR spectra of uMWCNTs show two strong absorption peaks at 3400 cm^−1^ and 1224 cm^−1^, which correspond to the stretching and bending vibration of –OH, respectively. The C=O and C–OH in –COOH exhibited the stretching vibrations at 1725 cm^−1^ and 1397 cm^−1^ [41,42,43]. The FTIR spectra illustrated that the unzipped MWCNTs had oxygen-containing functional groups formed at the oxidation by strong oxidants such as potassium permanganate, thus contributing to the interaction between the resin and the filler and to effective dispersing of the latter in the matrix.

Raman spectra (Figure 3b) demonstrates that the pristine MWCNTs shows a D band at 1346 cm^−1^, which is attributed to the defects on the surface or edge of uMWCNTs and the disordered structure of graphite in the MWCNTs, as well as a G band at 1574 cm^−1^ attributed to the C atom sp^2^ hybridized in-plane stretching vibration [44]. Compared with the MWCNTs, the shape and intensity of the 2D peak at 2700 cm^−1^ of uMWCNTs obviously changed. The value of I_D_/I_G_ (R) is often used to exhibit the degree of functionalization of MWCNTs [45]. The R value represented the degree of graphitization of carbon nanotubes. It can be seen from Figure 3b that the R value increases from 0.58 to 1.08 after unzipping, which reveals that a large number of defects [46] and functional groups are formed in the MWCNTs at unzipping, and the size of the in-plane C sp^2^ domains decreases, destroying the original ordered structure.

The XRD pattern of the MWCNTs shows the presence of a peak at 2θ = 25.9°, which is the corresponding graphite surface with spacing of (002) plane and corresponding d-spacing of 0.34 nm. After unzipping, the position of the main peak moved slightly to the position of 2θ = 24.7°, and the d-spacing became 0.36 nm. The peak of XRD spectrum at 2θ = 42.3° represented the characteristic diffraction peak of plane (100). This indicated that MWCNTs were unzipped to uMWCNTs and the most of the MWCNTs structure were changed. Due to the introduction of more oxygen-containing groups between adjacent layers, the d-spacing increases and the dense graphene layer falls off loosely.

To further investigate the changes of functional groups before and after the oxidation of MWCNTs, we characterized MWCNTs and uMWCNTs via XPS. The ratio of the content of carbon to the content of oxygen in XPS spectrum (O/C ratio) could characterize the degree of oxidation of carbon materials. The O/C ratio of MWCNTs and uMWCNTs is 0.0041 and 0.1593 respectively. The results show that the O/C ratios of uMWCNTs were higher than that of MWCNTs, indicating that the oxidation degree of uMWNTs increased. In the XPS spectra (Figure 4), the O content of MWCNTs is near zero, which can be considered to be caused by the moisture in the air. The peak intensity of O1s of uMWCNTs is much higher than that of MWCNTs, and can be divided into 532.3 eV C–O, 533.2 eV C=O. It proved that the surface of CNTs is modified with O.

In the uMWCNTs /Eu spectra, Eu^3+^ shows two peaks, namely Eu_3_d_5/2_ at about 1135 eV and Eu_3_d_3/2_ at about 1165 eV, which proved the existence of the Eu trivalent state. In addition, characteristic peaks of N1s and O1s were also observed in Figure 4e,f. According to the peak differentiation of O1s, there is an Eu–O peak near 533 eV, which indicated the existence of Eu in uMWCNTs/Eu and the formation of bonding with CNTs. The F and N elements in the sample also confirmed the existence of Eu(TTA)_3_phen in uMWCNTs/Eu. It can also be seen from the contents of each element in Table 1 that the contents of Eu, F and N elements in Eu(TTA)_3_phen in uMWCNTs/Eu are higher than those in MWCNTs/Eu, indicating that the content of europium complex in uMWCNTs/Eu is higher than that in MWCNTs/Eu. This is in connection with the increased surface area of uMWCNTs and the increase of groups on the surface and edge, which could adsorb more lanthanide metal complexes.

The luminescence properties of MWCNTs/Eu and uMWCNTs/Eu nanomaterials were characterized by ethanol dispersion. Figure 5a,b show the excitation spectra and emission spectra of MWCNTs/Eu and uMWCNTs/Eu. Figure 5a shows that the excitation wavelength of MWCNTs/Eu is 372 nm and that of uMWCNTs/Eu is 386 nm. Compared with MWCNTs/Eu, the excitation wavelength peak of uMWCNTs/Eu shifted to the right by 14 nm. It can be seen from the excitation spectra of MWCNTs/Eu and uMWCNTs/Eu that although the excitation characteristic peak positions of MWCNTs/Eu and uMWCNTs/Eu nanomaterials were similar. However, the peak position of uMWCNTs/Eu luminescent materials showed a red shift, indicating that the energy transfer occurred between the Eu(TTA)_3_phen and the uMWCNTs. This also indicated that Eu(TTA)_3_phen and uMWCNTs bond. In addition, it can be seen from Table 2 that although the luminescence lifetime of MWCNTs/Eu (424.1 μs) is lower than that of Eu(TTA)_3_phen (515.2 μs), the luminescence lifetime of uMWCNTs /Eu (606.1 μs) is higher than that of Eu(TTA)_3_phen. This indicates that the luminescence lifetime of the complex is improved after the Eu(TTA)_3_phen is combined with uMWCNTs.

### 3.2. The Strong Enhancement of CER Mechanical Properties by MWCNTs and uMWCNTs

As shown in Figure 6, the mechanical properties of CER network are enhanced strongly with the addition of MWCNTs or uMWCNTs. The flexural strength and tensile strength of the CER/MWCNTs nanocomposite were 120.75 MPa and 31.17 MPa, respectively, and increased by 33.0% and 24.4%, respectively, compared to the pure CER network. The effect of uMWCNTs is similar to the trend of CER/MWCNTs but the enhancement is more pronounced. It was found that the values of flexural strength and tensile strength of the CER/uMWCNTs nanocomposite were 160.07 MPa and 37.79 MPa, respectively, which are 32.6% and 21.3% higher than that of CER/MWCNTs nanocomposites, respectively. Compared with pure CER network, the flexural strength and tensile strength are increased by 76.3% and 50.9%, respectively.

The stress–strain curves for the pure CER network, CER/MWCNTs and CER/uMWCNTs nanocomposites are illustrated in Figure 7. The difference in mechanical properties observed was closely related to efficiency of chemical interaction of the MWCNTs and uMWCNTs with the CER matrix and dispersing nanofillers in the polymer. Compared with MWCNTs, the uMWCNTs had higher available interface area due to oxidative unzipping of MWCNTs. As shown in Figure 3a, the uMWCNTs possess many groups, such as –OH and –COOH groups, which react with the –O–C≡N group of CE monomer and growing CER network. Thus, this strong interfacial covalent bonding between uMWCNTs and the CER matrix (see Figure 1) results in more difficulty in sliding uMWCNTs in the CER matrix.

SEM images of tensile fracture of CER/uMWCNTs nanocomposites were also used to further characterize the dispersibility of uMWCNTs and possible strengthening mechanism. Figure 8 shows the scanning electron microscope (SEM) cross-sectional image of the nanocomposite. On the one hand, as shown in Figure 8b some agglomerations of MWCNTs in in the CER/MWCNTs nanocomposites are observed. The agglomerated regions of MWCNTs were easy to cause crack propagation. However, the dispersion of uMWCNTs in CER/uMWCNTs nanocomposites is more uniform (Figure 8c), which is conducive to the improvement of toughness. On the other hand, it can be seen that the MWCNTs have been pulled out of the matrix. Thus, we supposed the reinforcement effect is mostly pulling-out mechanism because the interaction force between MWCNTs and CER resin matrix was weak. However, the different situation can be found from the CER/uMWCNTs as shown in Figure 8c, in which the broken image of uMWCNTs is shown by the very short bright dots (Figure 8c). Obviously, the binding force between the uMWCNTs and the CER matrix was enhanced due to the increased surface area from unzipping procedure as well as the increased polar oxygen-contained groups at surface of uMWCNTs.

In addition, it was clear that the interfacial adhesion between uMWCNTs and CER matrix was provided not only by Van der Waals forces, but also by the chemical bonding that can bear higher loads. As shown in Appendix A., the characteristic peak of –OH group (3400 cm^−1^) in CER/uMWCNTs composites disappears. In addition, a new –C–N–C peak appears at 1566 cm^−1^. Therefore, this paper tends to react the –OH group and –COOH group on the surface of uMWCNTs with cyanate ester resin. Furthermore, as shown in Figure 1, the growing CER network grafted covalently to the surface of uMWCNTs via reaction of cyanate groups with –OH groups, forming the CER network coating on a surface of carbon nanotubes, and improving the binding force between carbon nanotubes and matrix polymer. Such a wrapping of CNTs by the CER was observed before [47], which was only possible in systems with high contact between the polymer and MWCNTs.

### 3.3. Thermal Property of CER/MWCNTs and CER/uMWCNTs Nanocomposites

The DSC measurements of CER/MWCNTs and CER/uMWCNTs nanocomposites were performed and the thermograms are given in Figure 9a. It can be seen that the MWCNTs or uMWCNTs were added to the CER monomer, the temperature positions of the peaks were 346.14 °C or 328.98 °C, respectively. This phenomenon proves that MWCNTs had a strong catalytic effect on CE polymerization and formation of CER network inside the nanocomposites [48]. At the same time, the melting enthalpy (ΔH_m_) of CER/MWCNTs was 84.87 J/g, while this value decreased to 78.02 J/g with the addition of uMWCNTs and the enthalpy of formation (ΔH_s_) of CER/MWCNTs and CER/uMWCNTs nanocomposites were 718.09 J/g or 692.82 J/g, respectively. The difference between uMWCNTs and MWCNTs was that the surface area of uMWCNT was larger, and there were many polar groups such as –OH group and –COOH group on the surface.

The thermal degradation behaviors of the CER/MWCNTs and CER/uMWCNTs nanocomposites are evaluated via TGA (Figure 9b). It can be seen that the degradation mechanism of CER/MWCNTs and CER/uMWCNTs is similar, but the thermal stability is obviously different. The initial decomposition temperature (T_di_) of the CER/MWCNTs nanocomposite was 338 °C, while the T_di_ of CER/uMWCNTs nanocomposite was 391 °C, which was 53 °C higher than that of CER/MWCNTs. The chemical grafting of uMWCNTs to the cyanate ester resin matrix improved the adhesion between uMWCNTs and CER matrix and effectively delayed the thermal decomposition of CER network, and thus enhanced the thermal stability of the nanocomposite. In addition, the char residue at 750 °C of CER/uMWCNTs nanocomposite was slightly higher than that of CER/MWCNTs nanocomposites, but the values were very close. One can concluded that a control of structure of the carbon nanofiller played a very important role of developing new nanocomposites with outstanding thermal property.

### 3.4. Aging Property of CER/MWCNTs and CER/uMWCNTs Nanocomposites under UV Irradiation

Some changes in polymer properties, such as mechanical properties, were associated with UV irradiation [49]. IR spectroscopy (Figure 10a) shows the effect of UV irradiation on structure of the composites. Figure 10a displays the FTIR of pure CER, CER/MWCNTs and CER/uMWCNTs nanocomposites before and after UV irradiation. CER network contained many triazine rings formed at polycyclotrimerization of cyanate monomer. The FTIR spectra of the pure cured CER and nanocomposites before and after UV irradiation shows that the peaks of triazine rings at 1570 cm^−1^ (–C–N=C–) and at 1370 cm^−1^ (–O–C=N–) remained unchanged. This obviously indicated that neither MWCNTs nor uMWCNTs could change the UV stability of CER, and thus the composites had also obtained excellent UV aging performance.

Figure 10b compares the mechanical properties of the pure CER network, CER/MWCNTs and CER/uMWCNTs nanocomposites before and after UV irradiation. After UV irradiation, the flexural strength of the material equaled 116, 133, 182 MPa, respectively that differed just about 2% from the values measured before UV irradiation. Similarly, the values of tensile strength for the pure CER network, CER/MWCNTs resin and CER/uMWCNTs nanocomposites remained unchangeable after UV irradiation (25, 32, 38 MPa, respectively). This result demonstrate that UV irradiation has little influence on the overall mechanical properties of CER. Since MWCNTs did not change the ultraviolet aging properties of the CER, the CER/uMWCNTs composites with both excellent mechanical properties and ultraviolet aging properties were obtained.

### 3.5. Luminescence Property of CER/MWCNTs/Eu, CER/uMWCNTs/Eu Nanocomposites

Photoluminescence spectra of CER/MWCNTs/Eu, CER/uMWCNTs/Eu nanocomposites were recorded. To improve the photostability of the complex, the complex was first doped with carbon nanotubes and then compounded into the resin matrix (see Section 2.2.1). Luminescence excitation and emission spectra of each component (CER, MWCNTs, uMWCNTs and Eu) of the composites are shown in Appendix A and Figure 11 shows the excitation and emission spectra of the cured CER doped with MWCNTs/Eu and uMWCNTs/Eu. After curing, the photoluminescence spectra of CER/MWCNTs/Eu, CER/uMWCNTs/Eu in Figure 11b distinguished luminescence peaks at 614 nm, indicating the emission characteristics of europium complexes according to the following transition ^5^D_0_ → ^7^F_2_ (614 nm); however, other transitions were concealed. On the other hand, with the formation of CER network in polymerization process, the luminescence emission spectrum of resin gradually appeared as a wide emission band around 500 nm, which indicated the emission of conjugate aromatic network. As shown in Figure 11b, the intensity of resin emission band even overwhelmed that at 614 nm from the europium complex. At this situation, the emission of europium complex became luminescence behind the visible (FBV), which could impart the embedment of photon quantum information into potential application target. Figure 11a,b indicates that the luminescence intensity of CER/uMWCNTs/Eu is higher than that of CER/MWCNTs/Eu with the same luminescence hybrid material content.

Figure 11c gives the luminescence decay curves of CER/MWCNTs/Eu, CER/uMWCNTs/Eu measured at the ^5^D_0_ → ^7^F_2_ transition of Eu^3+^ ion. The luminescence lifetime of CER/MWCNTs/Eu and CER/uMWCNTs/Eu is 8.61 μs and 186.39 μs, respectively (shown in Table 3). It can be seen that the addition of uMWCNTs improved luminescence stability of the CER. On the one hand, the surface area of uMWCNTs increased, and at the same time, a lot of functional groups were introduced, which could adsorb more lanthanide complexes. On the other hand, the stabilization effect of oxygen-containing groups such as –OH and –COOH groups with lanthanide rare earth complexes improved the energy transfer efficiency from ligand to Ln^3+^. Although carbon nanotubes could cause partial luminescence quenching, the luminescence quenching efficiency of original MWCNTs was higher than that of unzipped carbon nanotubes [50]. In conclusion, the luminescence intensity of CER/uMWCNTs/Eu is higher than that of CER/MWCNTs/Eu.

It can be seen from Table 3 that the measurement results of quantum yield are corresponding to the luminescence intensity and lifetime results, and one can see that the higher the quantum yield, the higher the luminescence intensity. This is also connected with the specific surface area and surface groups of uMWCNTs.

## 4. Conclusions

In this paper, the luminescence property was innovatively imparted to cyanate ester resin, which also combined with the original thermal properties and the improved mechanical property. With the addition of 0.1 wt.% of uMWCNTs, the flexural strength and tensile strength of composites increased 59.3% and 49.3%, respectively. With the same addition, the luminescence property injected into the cyanate resin matrix plays the role of emitting photon signals. The existing broad band of emission in blue makes the luminescence from MWCNTs/Eu an invisible photon signal, and provides the potential photon signal embedment. Another excellent property of the composites is its excellent UV aging performance; neither MWCNTs nor uMWCNTs could change the UV stability of CER. With these multiple functional and mechanical properties, it could be predicted that the thermal-resistant polymer with hybrid luminescent properties will be widely used in sensing in a high-temperature environment.

## Figures and Tables

**Figure 1 materials-14-04244-f001:**
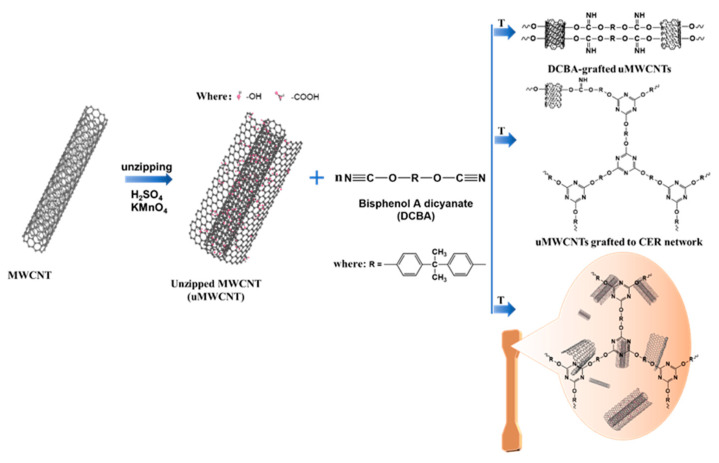
Schematic illustration of the preparation process used for CER/uMWCNTs nanocomposites.

**Figure 2 materials-14-04244-f002:**
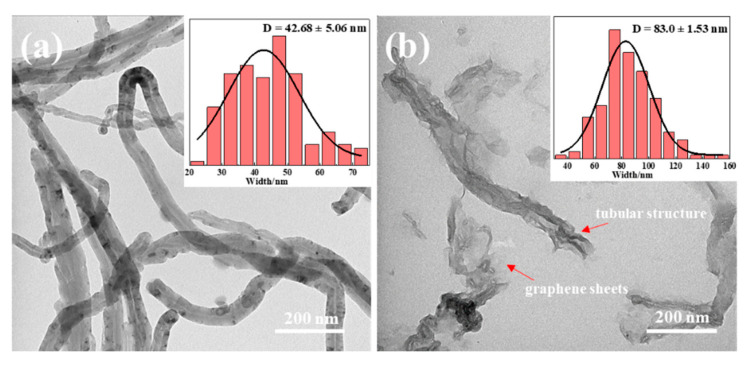
TEM images of (**a**) pristine MWCNTs, and (**b**) uMWCNTs.

**Figure 3 materials-14-04244-f003:**
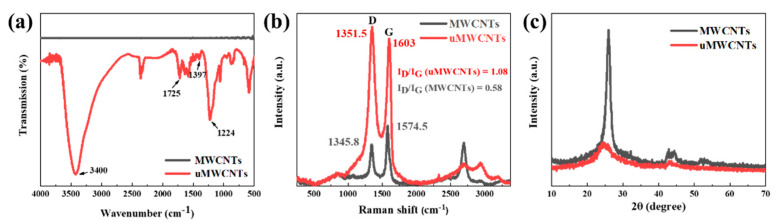
FTIR spectra (**a**), Raman spectra (**b**) and XRD patterns (**c**) of MWCNTs and uMWCNTs.

**Figure 4 materials-14-04244-f004:**
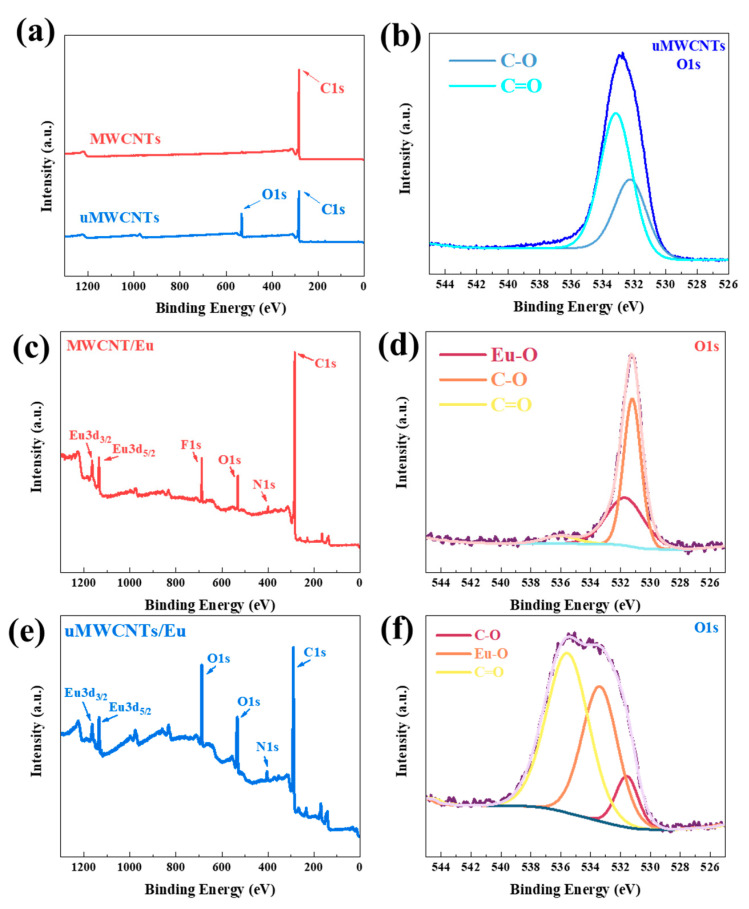
XPS spectra of MWCNTs (**a**); uMWCNTs (**b**,**c**); MWCNTs/Eu (**d**,**e**); uMWCNTs/Eu (**f**).

**Figure 5 materials-14-04244-f005:**
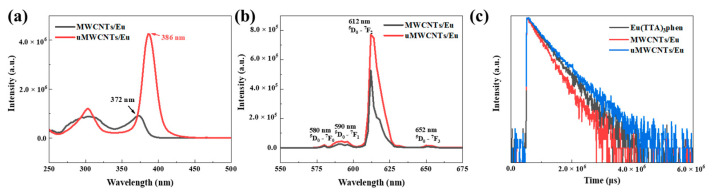
The excitation (**a**) and emission (**b**) spectra of MWCNTs/Eu and uMWCNTs/Eu and luminescence decay curves (**c**) of MWCNTs/Eu, uMWCNTs/Eu and Eu(TTA)_3_phen.

**Figure 6 materials-14-04244-f006:**
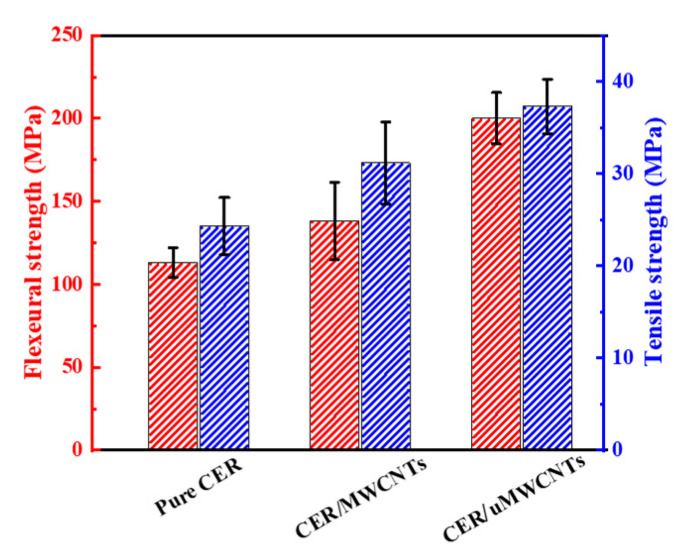
The mechanical properties of pure CER network, CER/MWCNTs and CER/uMWCNTs nanocomposites.

**Figure 7 materials-14-04244-f007:**
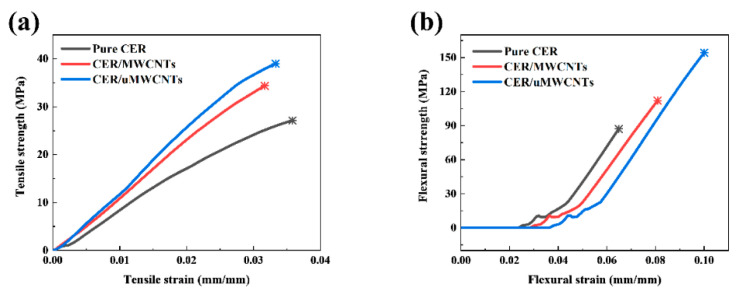
Tensile stress–strain curves (**a**) and flexural stress–strain curves (**b**) of the pure CER network, CER/MWCNTs and CER/uMWCNTs nanocomposites.

**Figure 8 materials-14-04244-f008:**
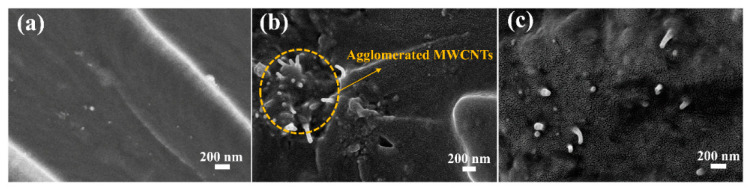
Cross-sectional SEM images for pure CER network (**a**), and for nanocomposites based on CER/MWCNTs (**b**) and CER/uMWCNTs (**c**).

**Figure 9 materials-14-04244-f009:**
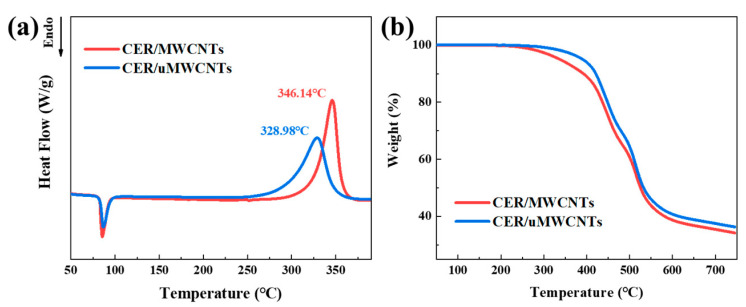
(**a**) DSC curves of polymerization of initial compositions (indicated in the plot); and (**b**) TGA curves for the CER/MWCNTs and CER/uMWCNTs nanocomposites.

**Figure 10 materials-14-04244-f010:**
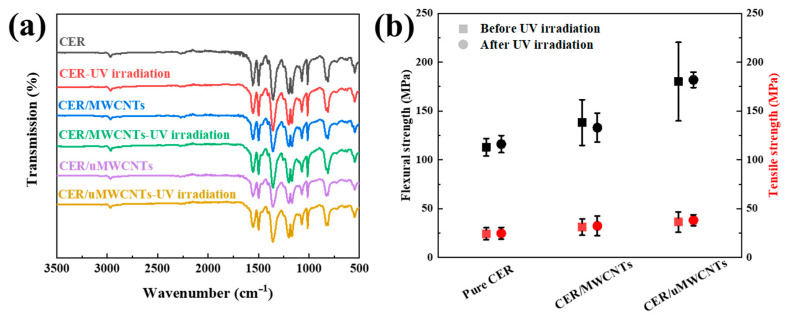
FTIR spectra (**a**), mechanical properties (**b**) of the pure CER network, CER/MWCNTs and CER/uMWCNTs nanocomposites before and after UV irradiation.

**Figure 11 materials-14-04244-f011:**
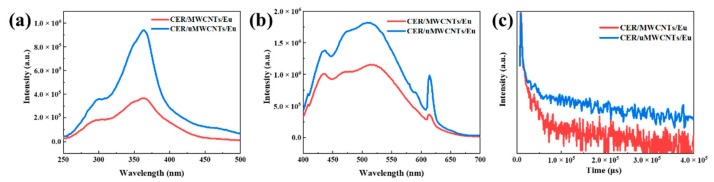
Luminescence excitation (**a**) and emission (**b**) spectra and luminescence decay curves (**c**) of CER/MWCNTs/Eu, CER/uMWCNTs/Eu nanocomposites. Inset shows changes Eu at λ_ex_ = 364 nm, λ_em_ = 614 nm, Scan Slit = 1.50.

**Table 1 materials-14-04244-t001:** XPS analysis results of MWCNTs/Eu, uMWCNTs/Eu, MWCNTs/Eu and uMWCNTs/Eu.

Sample	C (%)	O (%)	O/C	Eu (%)	F (%)	N (%)
MWCNTs	99.59	0.41	0.0041	–	–	–
uMWCNTs	86.32	13.75	0.1593	–	–	–
MWCNTs/Eu	82.67	7.44	0.0899	0.84	6.30	2.29
uMWCNTs/Eu	70.02	15.57	0.2223	1.30	9.33	4.24

**Table 2 materials-14-04244-t002:** Luminescence lifetimes of MWCNTs/Eu, uMWCNTs/Eu and Eu(TTA)_3_phen.

Sample	Lifetime (μs)
Eu(TTA)_3_phen	515.2
MWCNTs/Eu	424.1
uMWCNTs/Eu	606.1

**Table 3 materials-14-04244-t003:** The lifetime and quantum yield (QY) of CER/MWCNTs/Eu and CER/uMWCNTs/Eu nanocomposites.

Sample	Lifetime (μs)	QY (%)
CER/MWCNTs/Eu	8.61	1.02
CER/uMWCNTs/Eu	186.39	2.12

## Data Availability

The data presented in this study are available on request from the corresponding author after obtaining permission from an authorized person.

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
