# Peer review of "Improved Mechanical, Anti-UV Irradiation, and Imparted Luminescence Properties of Cyanate Ester Resin/Unzipped Multiwalled Carbon Nanotubes/Europium Nanocomposites"

_materials, 2021, doi:10.3390/ma14154244_

Round 1
Reviewer 1 Report
This paper deals aims to demonstrate that it is possible to prepare nanocomposites containing oxidized carbon nanotubes and europium luminscent complexes in a matrix of cyanate resin. However I've found several major concerns.
1) Authors talk about carbon nanotubes unzipping but their chemical process is a simple oxidation process. Authors claim that with their oxidation process, diameter of the nanotubes increases. This is completely wrong.
2) Authors state that there is some kind of chemical interaction between the tubes and the polymer matrix (see Figure 1). However, the surface of the oxidized tubes has not been characterized properly. XPS results are only used to find the O/C ratio; a detailed description of the functional groups must be given and it must be analyzed which kind of chemical reaction grafts the matrix to the surface of the tubes
3) Composites with europium complexes appear to be simple mixtures since there is no description of the interaction of the complexes either with the matrix or with the tubes
4) Authors assign the term "fluorescence" to the emission of Eu complexes. This is really strange due to the extremely large decays, in the order of microseconds, suggesting that it is a forbidden transition and not an emission from a singlet state.
Author Response
We sincerely thank you for the good questions. A point-by-point response to your comments is given below. We hope that we have successfully addressed the concerns.

Reviewer 2 Report
The article entitled “Improved Mechanical, Anti-UV Irradiation and Imparted Fluorescence Properties of Cyanate Ester Resin/Unzipped Multiwalled Carbon Nanotubes/Europium Nanocomposites” describes the CER/MWCNTs, CER/uMWCNTs as well as CER/MWCNTs/Eu, CER/uMWCNTs/Eu nanocomposites have been created and investigated on mechanical properties, heat resistance, UV aging resistance of the resin matrix and the fluorescence properties of the lanthanide complex.
The paper has minor revisions and therefore we recommend it is highly suitable for publication in Materials. The minor revisions are:
Introduction
Page no 2, Line 69-71 authors should add the suitable references to substantiate the claim made in that sentence.
Material and methods
- Page no 3, section 2.2.1, line 121-122 authors should explain the methodology used for washing.
- Page no 3, section 2.2.1, line 125-131 authors should elaborate the methodology such as composition of the materials, what is the requirement for the pH 6.
Results and discussion
Page no 5, Line 203-204, authors should check the spelling and throughout the article.
Author Response
We are so grateful for the reviewer’s encouraging comments and recommend it is highly suitable for publication in Materials. Our point-by-point responses to the reviewer’ comments are detailed below.

Reviewer 3 Report
The paper under consideration reports the improvement of various properties of cyanate ester resin (CER) after doping with unzipped multiwalled carbon nanotubes (uMWCNTs). The authors performed a rigorous analysis of the resin nanocomposites' structural, mechanical, and thermal properties. They also demonstrated the effect of the nanocomposite resin matrix on the photoluminescence of embedded Eu3+ ions. The work is done well, the obtained results look reliable and the manuscript is well written. Therefore, the paper can be accepted for publication after the minor corrections as following.
- The authors have performed a study of UV-induced aging of the prepared nanocomposites and concluded that their “ultraviolet aging performance is excellent” (p. 13, lines 452-455). However, this conclusion is not supported by the experiments. Indeed, the FTIR spectra measured before and after the UV irradiation (Figure 10a) show no changes for CER/MWCNTs and CER/uMWCNTs nanocomposites. However, no spectral changes are demonstrated for pure CER as well. Another set of data related to mechanical properties of the UV-aged nanocomposites (Figure 10b) also shows no effect of UV on neither nanocomposites nor pure resin within the experimental error. Taking into account the time of the UV exposure (40 days), one can conclude that MWCNTs (both zipped and unzipped) do not change the UV stability of CER. Therefore, the conclusion in the paper is incorrect in the current form and should be rewritten.
- The authors give a detailed description of all stages of the samples preparation and their characterization. However, the outcome of the unzipped MWCNTs remained hidden. How is effective the unzipping procedure? What is the ratio of unzipped NTs in the resulting powder? Since this powder is used for the creation of a composite, the level of its purity is crucial for understanding the reliability of the reported results.
- The MWCNTs unzipping naturally leads to an increase in the characteristic width of the NTs, and the authors demonstrated this with TEM images. Indeed, the histograms in Figure 2 show twice larger width for uMWCNTs than that of pristine MWCNTs. However, the TEM images themselves do not corroborate this statistic. It is clearly seen that the width of the largest pieces of NTs in Figure 2b are the same or less than those in Figure 2a, whereas the scale bar is the same for both images. So, there is, probably, some mistake in these images. An additional recommendation is to increase the font size of the histograms. In the current form, it is hard to read the numbers even with 600% page magnification.
- The authors report data that are not discussed in the text. Table 2 contains information about the fluorescent lifetimes of Eu-containing compounds, but these data are not discussed in the text at all. A similar situation occurs with Table 4. Figure 8c and Tables 2, 3, and 4 are not cited in the text. If they are not important, it would be better to remove them from the manuscript.
- The manuscript has only a few misprints:
- Figure 4e,g. XPS peaks corresponding to F are not labeled.
- P. 2, lines 94-95. No need to insert the abbreviation for the second time.
- P. 5, line 191. Written “the length of MWCNTs”, should be “the length of uMWCNTs”
- P. 8, line 281. Written “indicated that Eu(TTA)3phen”, should be “indicated that Eu(TTA)3phen”
- P. 10, line 358. Written “while the Tdi of…”, should be “while the Tdi of…”
- P. 11, line 407. Written “in Figure 9b”, should be “in Figure 11b”
To conclude, the manuscript under consideration is well written, the reported study is designed correctly, and the results look reliable. The paper can be accepted to the Materials journal after a minor revision.
Author Response
We sincerely thank the reviewer for the fruitful and understandable review provided for our manuscript. We appreciate that the reviewer's opinion that “The work is done well, the obtained results look reliable and the manuscript is well written”. Our point-by-point responses to the reviewer’s comments are detailed below.

Reviewer 4 Report
This work is interesting and the manuscript was well prepared. However, there are some problems needed to be addressed before publishing the article.
- The novelty of the work is not clear. The authors should clearly mention it.
- The first two sentences of the abstract should be rephrased to clarify how many novel composites are prepared and what made them significant or important.
- The authors should highlight the application of their findings both in abstract and conclusion section of the manuscript.
- The Materials and Methods section should include more references. The authors should provide references to support the methods which are used in this work.
- Figure 9 should include the curves of only MWCN and uMWCN as control.
- Figure 11 should include the curves of each of the components (CER, MWCN, uMWCN and Eu) of the composites as control. The concentrations of the components in the solution should be equivalent to their proportions in the composite. The curves may be given in the supplementary material.
Author Response
We sincerely thank the reviewer for the fruitful and understandable review provided for our manuscript. We appreciate that the reviewer's opinion that “This work is interesting and the manuscript was well prepared”. Our point-by-point responses to the reviewer’s comments are detailed below.

Round 2
Reviewer 1 Report
Authors have properly addressed all the concerns .